# Inflammatory Profile of Different Absorbable Membranes Used for Bone Regeneration: An In Vivo Study

**DOI:** 10.3390/biomimetics9070431

**Published:** 2024-07-16

**Authors:** Vinícius Ferreira Bizelli, Arthur Henrique Alécio Viotto, Izabela Fornazari Delamura, Ana Maira Pereira Baggio, Edith Umasi Ramos, Leonardo Perez Faverani, Ana Paula Farnezi Bassi

**Affiliations:** Department of Diagnosis and Surgery, School of Dentistry, São Paulo State University, UNESP, Araçatuba 16015-050, SP, Brazil; arthur.viotto@unesp.br (A.H.A.V.); izabela.delamura@unesp.br (I.F.D.); ana.baggio@unesp.br (A.M.P.B.); yassed_20@hotmail.com (E.U.R.); leonardo.faverani@unesp.br (L.P.F.); ana.bassi@unesp.br (A.P.F.B.)

**Keywords:** biomaterials, angiogenesis, bone regeneration

## Abstract

Background: Guided bone regeneration (GBR) has become a necessary practice in implantology. Absorbable membranes have shown advantages over non-absorbable membranes, such as blood support of bone tissue. This study aimed to evaluate five collagen membranes in rat calvaria critical-size defects through a histomorphometric analysis of the inflammatory profile during the initial phase of bone repair. Materials and methods: A total of 72 Albinus Wistar rats were used for the study, divided into six groups, with 12 animals per group, and two experimental periods, 7 and 15 days. The groups were as follows: the CG (clot), BG (Bio-Gide^®^), JS (Jason^®^), CS (Collprotect^®^), GD (GemDerm^®^), and GDF (GemDerm Flex^®^). Results: Data showed that the BG group demonstrated an inflammatory profile with an ideal number of inflammatory cells and blood vessels, indicating a statistically significant difference between the JS and CS groups and the BG group in terms of the number of inflammatory cells and a statistically significant difference between the JS and CS groups and the GD group in terms of angiogenesis (*p* < 0.05). Conclusions: We conclude that different origins and ways of obtaining them, as well as the thickness of the membrane, can interfere with the biological response of the material.

## 1. Introduction

Bone tissue is a mineralized connective tissue comprising organic (cells and extracellular matrix) and inorganic (mineral) components that fracture upon receiving a functional load that exceeds its elastic modulus. The process of healing is complex, multistep, and highly sensitive to mechanical signaling, comprising the stages of inflammation, repair, and remodeling, which is similar to other tissues; however, what differentiates bone tissue is its capacity to regenerate completely [1]. Any change or imbalance in one of these phases can lead to the failure of this process [2].

Due to the high rates of complications and challenges in performing surgical techniques with non-absorbable membranes [3,4,5], absorbable membranes have gained popularity in grafting procedures due to their high success rate and avoidance of a second surgery for their removal [6,7,8]. Materials such as collagen derived from various animal sources, bovine cortical bone, and aliphatic synthetic polyesters, as well as associations with bone proteins or biomaterials, have been used to enhance GBR [6,9,10].

Understanding that the initial inflammation caused by surgical trauma is one of the phases of bone repair or regeneration, the presence of external factors in GBR causes additional inflammation, thereby affecting the repair dynamics [11]. Thus, biocompatible materials, which attenuate this process and facilitate the vascularization of the tissue to be regenerated, is an important characteristic of the membranes linked in this process. Collagen, the raw material most used in the production of absorbable membranes, undergoes biological degradation that can affect the repair process in different proportions depending on its processing method, which can alter the tissue response negatively or positively in terms of the formation of bone islands, rather than new bone connected to the host bone [12,13,14,15,16,17]. In vivo and in vitro studies are still needed to study the biological behavior of all absorbable membranes that are composed of collagen.

The knowledge and control of the first phase, the inflammation phase, can help to explain the bad results found in clinical practice; so, studies have been paying attention to this phase. It is well known that membranes act as physical barriers for the GBR procedures and there are many differences between non-absorbable and absorbable membranes as well as their origins [4,5,12,13,14].

Absorbable membranes are composed of type I and III collagen or both in combination. Collagen can promote cell adhesion and proliferation, allowing bone formation to act as a carrier for various bioactive molecules. However, so that they can have more durability and structural resistance, these membranes can be cross-linked, which can interfere with biocompatibility [18,19,20]. Therefore, this study aimed to evaluate the initial biological behavior during the inflammatory phase of five commercially available absorbable membranes [14,15,16,17,18,19,20].

## 2. Materials and Methods

### 2.1. Samples

This research was approved by the Ethical Committee in the Use of Animals of the Araçatuba Dental School under the protocol 00217-2016, in accordance with the Animals in Research: Reporting In Vivo Experiments (ARRIVE) Guidelines.

Seventy two male, adult (aged, 3–4 months) rats (Rattus *norvegicus* albinus, Wistar), weighing approximately 250–350 g, were used in this study. These rats were randomly divided into six groups (*n* = 6 per group) and the experimental durations were 7 days and 15 days after surgery. These animals were kept in cages blindly identified to the operator and fed a balanced ration (1.4% calcium and 0.8% phosphorus; NUVILAB, Curitiba, PR, Brazil) and water ad libitum with day/night cycles at the vivarium of the university. An 8 m critical-size defect was performed in all animals and filled with clot (CG—negative control), Bio-Gide^®^ (BG), Jason^®^ (JS), Collprotect^®^ (CS), GemDerm^®^ (GD), and GemDerm Flex^®^ (GDF). For this, we used a published study as reference to run the power test and decide the sample size, which was decided as 5 animals per group [20].

### 2.2. Experimental Surgery

Fasting was performed and the animals were prepared with intramuscular sedation by an intramuscular administration of ketamine hydrochloride (Francotar; Vibrac do Brasil Ltd.a, São Paulo, Brazil), along with xylazine (Rompum, Bayer AS, Saúde Animal, São Paulo, Brazil), at dosages of 0.7 mL/Kg and 0.3 mL/Kg, respectively.

The surgical principles of preparing the region with calvaria trichotomy, aseptic protocol, and sterilization of instruments and drapes were followed, ensuring biosafety during the procedures. After this, the flap was created, exposing the entire calvaria region so that the defect could be created and treated according to each group. The animals were medicated with a single dose of 0.2 mL of penicillin G benzathine (Pentabiotic Veterinário Pequeno, Fort Dodge Saúde Animal Ltd.a., Campinas, SP, Brazil) and monitored every 2 days post-operatively.

### 2.3. Histological Analysis

All samples obtained after euthanasia were removed and fixed in 10% formaldehyde solution for 48 h, washed in running water for 24 h, decalcified in 20% EDTA for 5 weeks, dehydrated sequentially in alcohol, and cleared. Subsequently, the bone defect was longitudinally separated in half. The pieces obtained were embedded in paraffin and received semi-serial cuts of 6 μm thickness. Six slides were obtained from each piece, which were stained with H&E, following which histological analysis was performed. These samples were coded and a single-blinded, calibrated examiner performed the analysis.

### 2.4. Inflammatory Profile

For the inflammatory profile (primary outcome) by the cell and blood vessel count, two sections were photographed at 100× magnification with a light microscope (DM4000B, Leica), a color image processor (Leica Qwin V3 software, Leica, Wetzlar, Germany), one color camera (DFC 500, Leica), and a computer (Intel Core I5, Intel Corp., Santa Clara, CA, USA; Windows 10, Microsoft Corp., Redmond, WA, USA), with the first image taken in the center of the defect just below the interface between the membrane and tissue, followed by one to the right and one to the left, resulting in a total of 36 images per experimental time per group. In the ImageJ program, a grid with 130 points was determined, and each cell (lymphocytes) or set of vessels that touched the intersections of the points were counted.

### 2.5. Statistical Analysis

All tests were performed using the Sigma Plot 12.3 statistical program (Systat Software, Inc., San Jose, CA, USA). Initially, the data obtained from the histometric analysis were subjected to a normality test (Shapiro–Wilk, *p* > 0.05). After confirming the normal distribution of the data, the two-factor analysis of variance and Tukey’s post hoc tests were used to compare means with a significance level of *p* < 0.05.

## 3. Results

There were no trans- or post-operative complications among the samples, allowing the inclusion of all of them in this study.

The results obtained were interpreted to observe the inflammatory profile of each membrane tested in comparison with the positive control (BG). Statistically significant differences were identified considering the number of inflammatory cells and blood vessels when comparing between all groups (*p* = 0.017 and *p* = 0.001, respectively).

When analyzing the period from 7 to 15 days in an intragroup comparison, a statistically significant difference was observed in the BG (*p* = 0.027), JS (*p* < 0.001), CS (*p* < 0.001), and GD (*p* = 0.002) groups, while the GDF group (*p* = 0.980) showed no difference, maintaining the inflammatory response at practically the same cellular level (Figure 1). In the comparative analysis between groups in the 7-day period, the BG group had the lowest inflammatory cell content, with statistically significant differences in the JS (*p* < 0.001) and CS (*p* = 0.024) groups. The GDF group had the second lowest cellular expression, showing a statistical difference for the JS (*p* < 0.001) and CS (*p* = 0.015) groups. The JS group, in addition to presenting the greatest amount of cellular content, also presented a statistically significant difference compared with the GD group (*p* = 0.010), which had an intermediate performance. At 15 days, the JS group demonstrated a mild inflammatory response with statistical differences compared with the BG (*p* = 0.021) and GD (*p* = 0.020) groups. The BG, CS, and GD groups demonstrated a smaller number of components in the inflammatory process, and the GDF group maintained cellular activity at the same level (Figure 2).

Despite the angiogenic property of the membranes, in an intragroup comparative analysis, the BG (*p* = 0.029) and GDF (*p* < 0.001) groups showed a significant difference from 7 to 15 days. The other groups showed a decrease in the number of blood vessels (Figure 3).

At 7 days, in the intergroup analysis, the BG and JS presented similar data and, between the CS, GD, and GDF groups, no statistically significant difference was demonstrated. Moving onto the second postoperative period, the BG and GDF groups showed statistical differences for the other three groups: CS, GD (*p* < 0.001), and JS (*p* = 0.008) (*p* = 0.009), respectively. The GDF group showed the greatest angiogenic potential, followed by the BG group. The CS group demonstrated a higher decrease in blood vessel number (Figure 4).

With the proposal to perform the intergroup analysis depending on the origin of the membranes, two additional analyses were performed: one only for those derived from collagen and the other for those derived from bovine cortical bone.

Among the collagen membranes analyzed, in the comparative analysis of the membrane factor and in terms of the cell count, statistical differences were observed in the number of inflammatory cells between the control group BG and the two test groups [CS (*p* < 0.029) and JS (*p* < 0.001)] and, in the quantification of blood vessels, statistical differences were observed only between the BG and CS groups (*p* < 0.001) (Figure 5).

Among the bovine cortical bone membranes analyzed, in the comparative analysis of the membrane factor, no group showed a statistical difference in terms of the number of lymphocytes. However, in the quantification of blood vessels, a statistically significant difference was observed between the BG and GD (*p* < 0.001) and the GDF and GD (*p* = 0.012) groups, and the BG and GDF groups had similar performances (Figure 6).

The photomicrographs, as shown in Figure 2, demonstrated that the membranes had distinct inflammatory profiles. The BG porcine collagen membrane represented by the BG group was used as a reference owing to its success, which has already been well elucidated in the literature. We realized that the ideal membrane profile would be a significant decrease in the number of inflammatory cells in the experimental period from 7 days to 15 days, accompanied by a significant increase in the number of blood vessels in the same period. The JS group presented different biological behavior; in the 7-day period, this caused a very large inflammatory infiltration. The performance of the CS group was compromised by the large number of cells observed at 7 days and by the decrease in the number of blood vessels at 15 days, suggesting that the membrane proved to be ineffective in vascular neoformation. Similar findings were observed in the JS group. The GD group showed a large decrease in the number of cells during the repair process and angiogenesis was not as significant. The GDF group had a peculiar behavior in relation to the inflammatory cells, maintaining the same level of its cellular activity profile; however, it showed a significant increase in the number of blood vessels, which provided a more balanced inflammatory response (Figure 7 and Figure 8).

## 4. Discussion

There are certain raw materials from which we can produce absorbable membranes, such as collagen, the substrate most used by several companies, and polymers. The comparison of collagen membranes using only the information reported in the product catalogs, which mostly refers to different processing conditions, sizes, and physical properties, is insufficient in determining which membrane delivers the best results and their precise clinical applications, making it challenging for dental surgeons to choose the best product to use in their procedures. This study, despite having used an animal model that promotes limitations such as membrane collapse, generated interesting data and was compatible with others found in the literature.

Thus, like any other material developed via engineering with a focus on healthcare, experimental work is needed to verify whether these materials meet the requirements for clinical application [21,22,23,24]. Thus, understanding the inflammatory profile promoted by various absorbable physical barriers allows us to understand how they act in the initial days of the repair process and why certain membranes showed more favorable results in GBR. Thus, in this work, we have absorbable membranes composed of collagen, which were analyzed in the initial process of bone neoformation.

Collagen membranes must be biotolerant and bioactive to interact with the organism, contribute to neovascularization, and promote mechanical support. The blood supply is necessary for nutrients and cellular components to reach the site of interest, the bone defect, thus ensuring the synthesis of the bone matrix by the osteoblast, so a strong mechanical surface is necessary to maintain blood clot stability [25,26,27,28].

Collagen, coming from various sources, provides membranes with the ability to allow the exchange of nutrients between the biomaterial and adjacent tissues, while ensuring the characteristic of cell selection. The great disadvantage of collagen is the lack of mechanical support, which culminates in collapse. Therefore, to improve this property, companies use molecular crosslinking techniques, mineralization, and multilayer membranes, which cause the inflammatory response to be more exacerbated and may interfere with the biocompatibility [29,30].

Regarding BG, it was used because of the high number of papers that describe its performance; these are considered a reference when considering absorbable barriers to GBR [31,32]. In this study, this performance can be attributed to the excellent capacity of neovascularization with large blood vessels and a low inflammatory response at 7 and 15 days with a considerable reduction in inflammatory cells from one period to the other. Therefore, the double layer does not interfere with biocompatibility [11,29].

The JS membrane, when compared to BG, has a structure with intertwined fibers of differently oriented collagen fibers; however, in this study, it promoted many inflammatory cells, especially leukocytes. The type of laboratorial treatment for collagen stabilization may have contributed to the lower blood vessel formation observed in the JS group at 15 days compared to the BG group [3,30]. Thus, the degradation of the porcine pericardial membrane caused a greater production of inflammatory cells than that of the porcine collagen membrane, which could negatively affect the final volume of the newly formed bone tissue.

The other tested membrane derived from the porcine dermis collagen CS group had a porous structure allowing the development of a large inflammatory infiltrate and a low capacity for vascular neoformation. During collagen manufacturing, chemical changes can modify its natural properties, making it biocompatible and suitable for GBR [33]. The type of membranes and their characteristics were summarized in the Table 1.

Regarding the results of the CS group compared with that of the BG group, significant differences were observed. At 7 days, the group presented more inflammatory cells associated with statistical difference, and the same occurred at 15 days, even though no statistical difference was observed in this period. According to some authors, the degradation of collagen and components used in fabricating membranes can influence the repair process and inflammatory reaction; this is why several materials are studied with the aim of improving clinical performance and maintaining the necessary biological properties [34,35]. However, other research has demonstrated excellent results when using absorbable membranes associated with biomaterials [36]. Similar findings were observed when vessels were analyzed at 7 days and 15 days; fewer vessels were formed in the CS group than in the BG group, although the thickness of the CS membrane was smaller than that of the BG membrane. In the intragroup analysis, the BG group was capable of forming more blood vessels, while the CS group demonstrated a decrease. Since angiogenesis is essential for the formation of bone tissue in GBR processes, the CS group may have lower bone neoformation than the BG group [18].

Bovine cortical membranes GD and GDF contain pores along the entire membrane. According to some authors, the structured collagen bone matrix serves as an ideal substrate for the recruitment and anchorage of osteoprogenitor cells, thereby facilitating the proliferation and differentiation of osteoblasts and acting as a transporter and protector of bone morphogenetic proteins against non-specific proteolytic enzymes [36,37,38,39]. They are considered compatible biomaterials, as evaluated by Bernabé et al., who concluded that the membrane derived from demineralized bovine cortical bone is well tolerated by tissues and is completely reabsorbed after 30–60 days by mononuclear cells and multinucleated giant cells, which disappear at the end of the process [40,41]. The membrane derived from bovine cortical bone remains functional until the final analysis period, as it is resorbed more gradually than other collagen membranes and porcine collagen membranes can increase the production of proinflammatory mediators [38,40,41,42]. This maintenance for a longer period is probably related to the cross-links naturally present in the material, suggesting that the inclusion of artificial cross-links to increase their resistance to resorption is not necessary. The same was reported by Rothamel et al. (2005) [29].

These characteristics demonstrate that these membranes have good potential for osteopromotion, as verified by Bassi et al. (2020) [10]. In this study, it was possible to verify their good biological behavior, especially that of GDF, which presented several inflammatory cells, like BG, at the beginning of GBR, although it did not decrease at 15 days. In terms of the vascular aspect, it also had a behavior very similar to that of BG, allowing a significant increase in vascularization at 15 days. In relation to GD, this study presented a biological behavior that was slightly inferior to that of BG and GDF, thus demonstrating that the membrane thickness had a direct relationship with the performance of the membrane, a finding that was corroborated by Bassi et al. (2020) [10].

This study addresses the initial phase of the repair process, focusing on how absorbable membranes behave biologically. The animal model used allowed us to perform a critical defect so that the biological potential was maximized; however, this same model offered some limitations, such as the possible collapse of the membrane and the use of different animals in each experimental period. Although the methodology presented has already received scientific proof, other in vitro analyses may contribute to future results.

## 5. Conclusions

Thus, it is concluded that collagen membranes of porcine origin have large variables, and the way the product is manufactured directly influences its quality and performance. This does not seem to occur with bovine cortical membranes that have more collagen stability, influenced in this case by its thickness and not by its constitution.

## Figures and Tables

**Figure 1 biomimetics-09-00431-f001:**
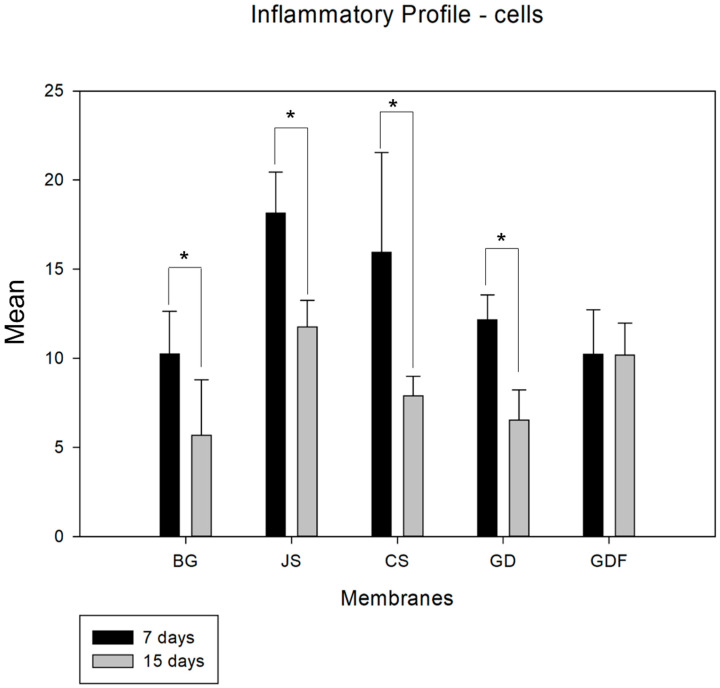
Graph demonstrating the intragroup comparison for the analysis of inflammatory cells (lymphocytes) in the 7-day and 15-day periods. Statistically significant differences were observed in the BG (*p* = 0.027), JS (*p* < 0.001), CS (*p* < 0.001), and GD (*p* = 0.002) groups. * demonstrates statistical differences (*p* < 0.05).

**Figure 2 biomimetics-09-00431-f002:**
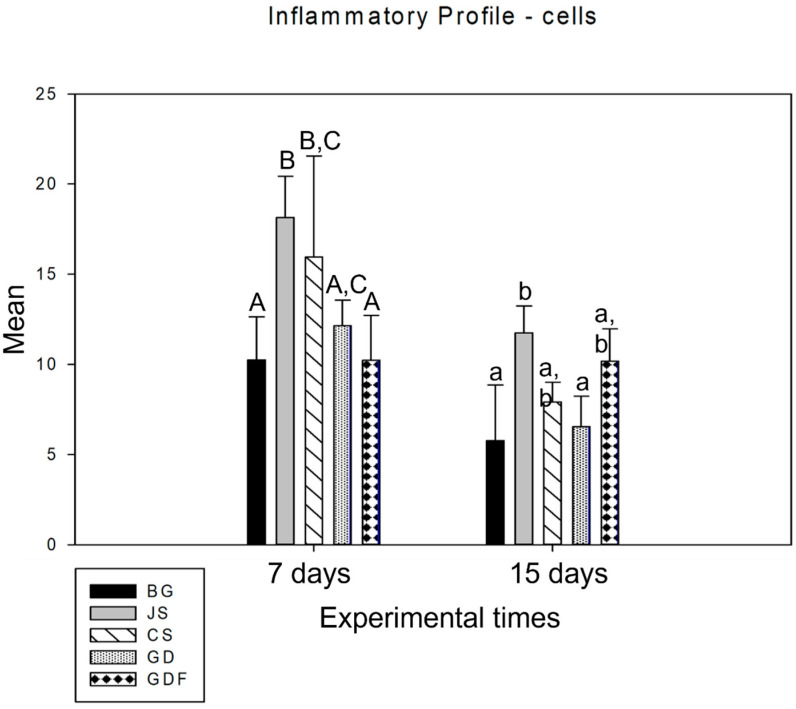
Graph showing the intergroup comparison over the 7-day and 15-day periods. The statistical differences observed at 7 days were between the BG and JS (*p* < 0.001), BG and CS (*p* = 0.024), JS and GD (*p* = 0.010), JS and GDF (*p* < 0.001), and CS and GDF groups (*p* = 0.015), represented by capital letters. JS and BG (*p* = 0.021) and JS and GD (*p* = 0.020) at 15 days are represented by lowercase letters.

**Figure 3 biomimetics-09-00431-f003:**
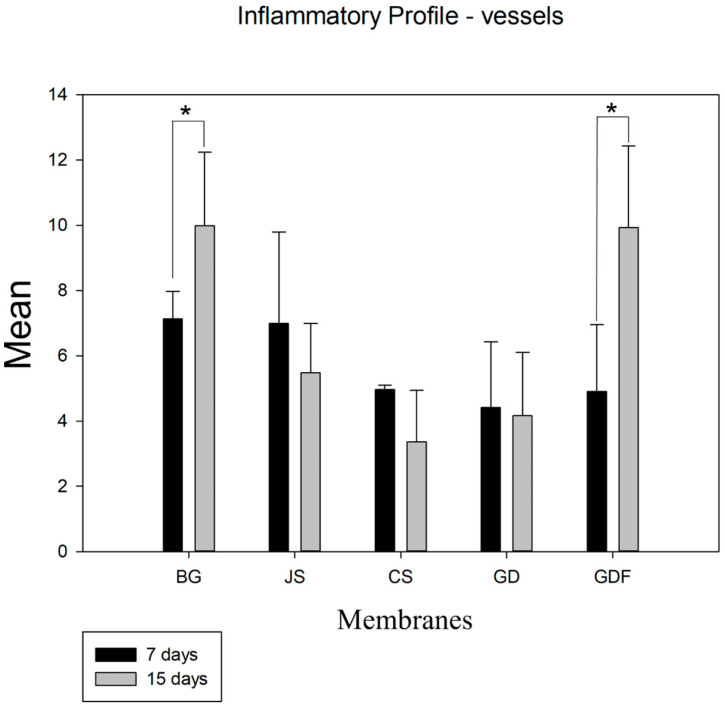
Graph demonstrating the intragroup comparison for blood vessel counts in the 7-day and 15-day periods. Statistically significant differences were observed only in the BG (*p* = 0.029) and GDF (*p* < 0.001) groups. * demonstrates statistical differences (*p* < 0.05).

**Figure 4 biomimetics-09-00431-f004:**
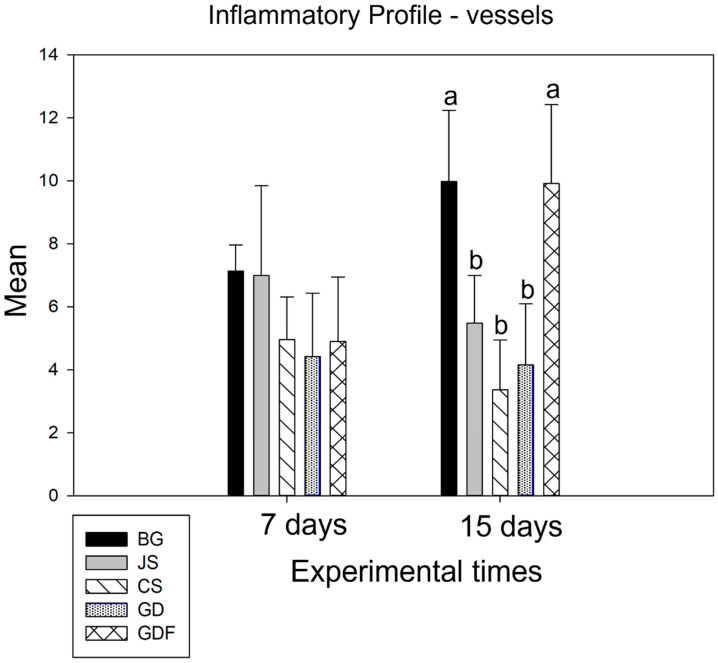
Graph demonstrating the intergroup comparison in the 7-day and 15-day periods. No statistical difference was observed at 7 days. At 15 days, the BG and GDF groups showed statistical differences compared with the other three groups: CS, GD (*p* < 0.001), and JS (*p* = 0.008) (*p* = 0.009), respectively, represented by lowercase letters.

**Figure 5 biomimetics-09-00431-f005:**
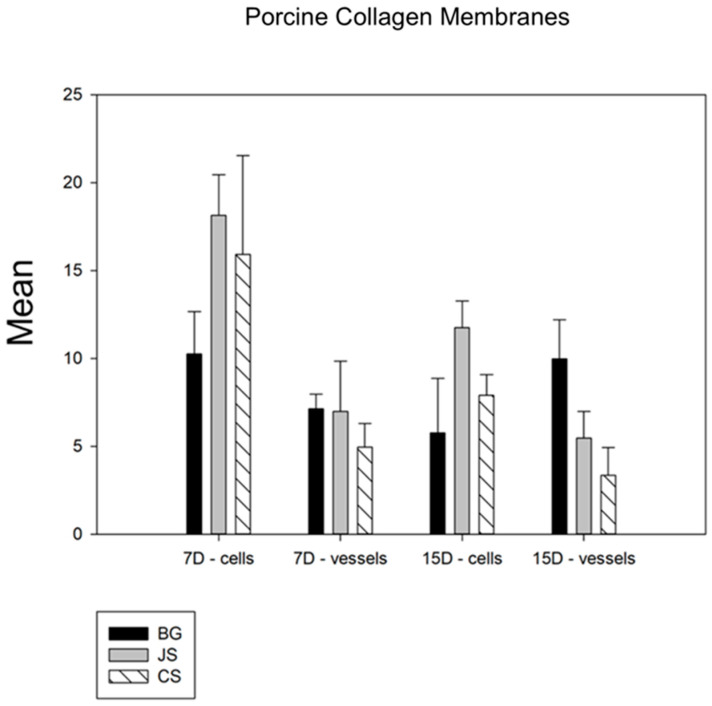
Comparative graph between the membranes derived from collagen. The BG group showed statistical differences compared with the two test groups, with the JS group showing a slightly better performance than the CS group. Thus, BG > JS > CS.

**Figure 6 biomimetics-09-00431-f006:**
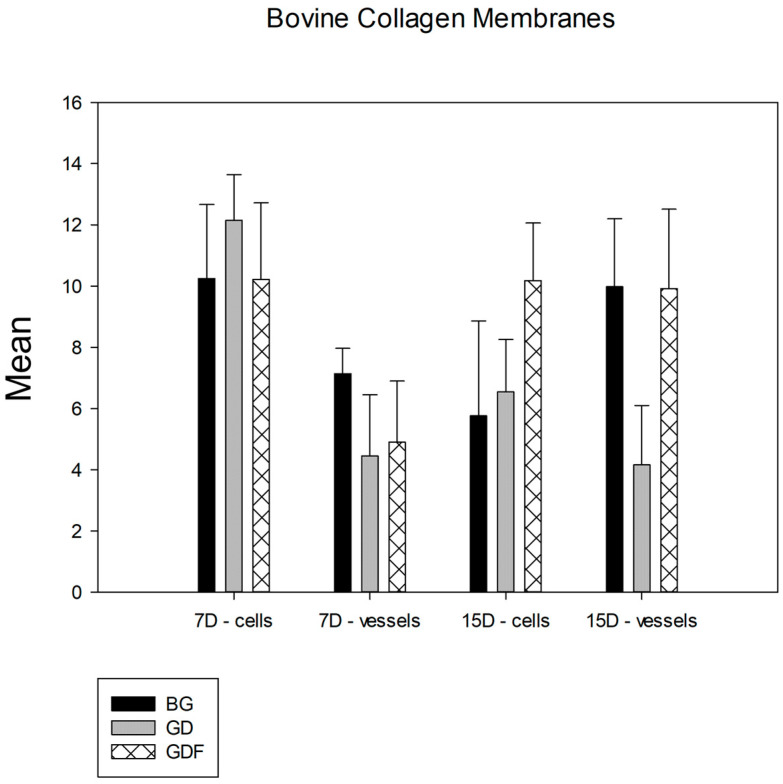
Comparative graph between bovine cortical bone membranes. The groups showed statistical differences only in terms of the angiogenic property, and the GDF test group had a behavior similar to that of the BG control group. Thus, BG = GDF > GD.

**Figure 7 biomimetics-09-00431-f007:**
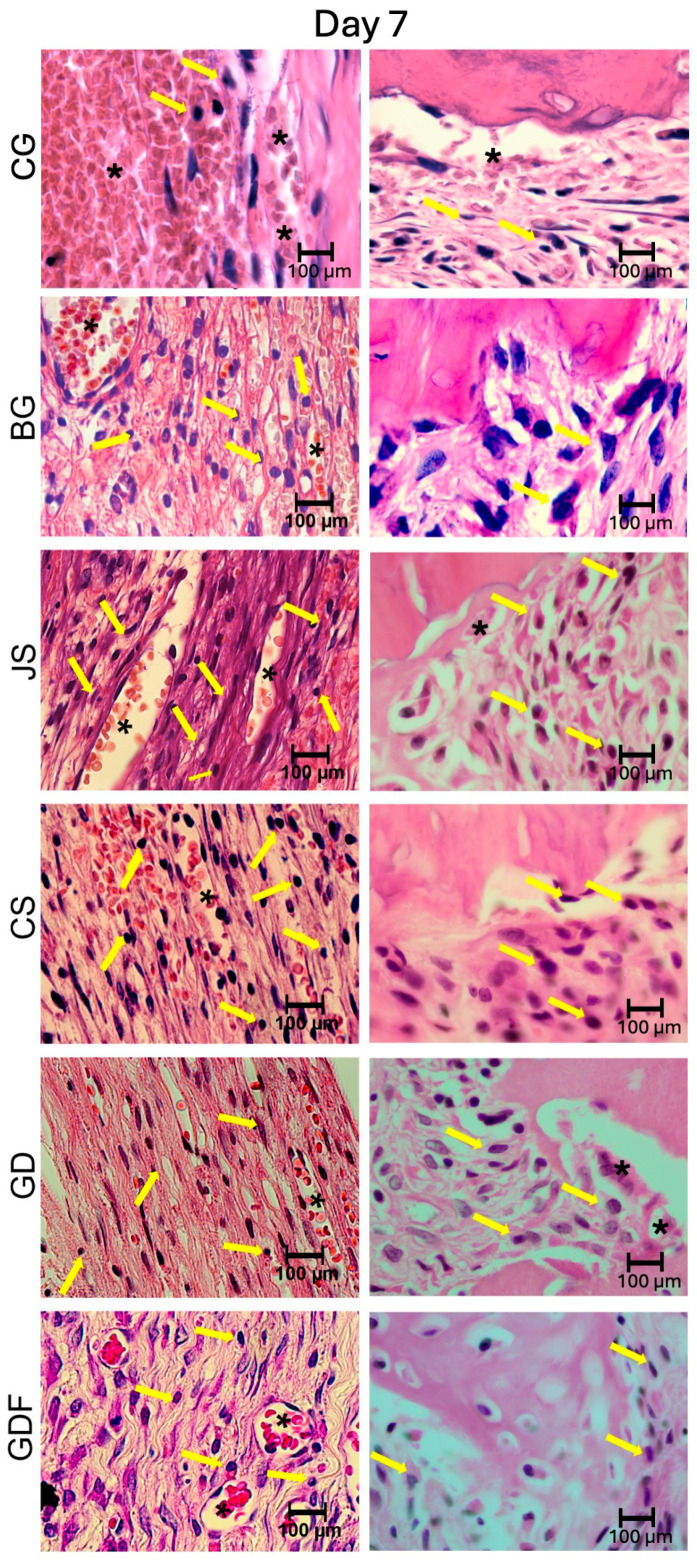
Photomicrographs at 100× magnification demonstrating the presence of cellular content (inflammatory cells and blood vessels) in the CG, BG, JS, CS, GD, and GDF groups at 7 days at the edges and at the center of the defect. Lymphocytes are indicated by yellow arrows and blood vessels by black asterisks.

**Figure 8 biomimetics-09-00431-f008:**
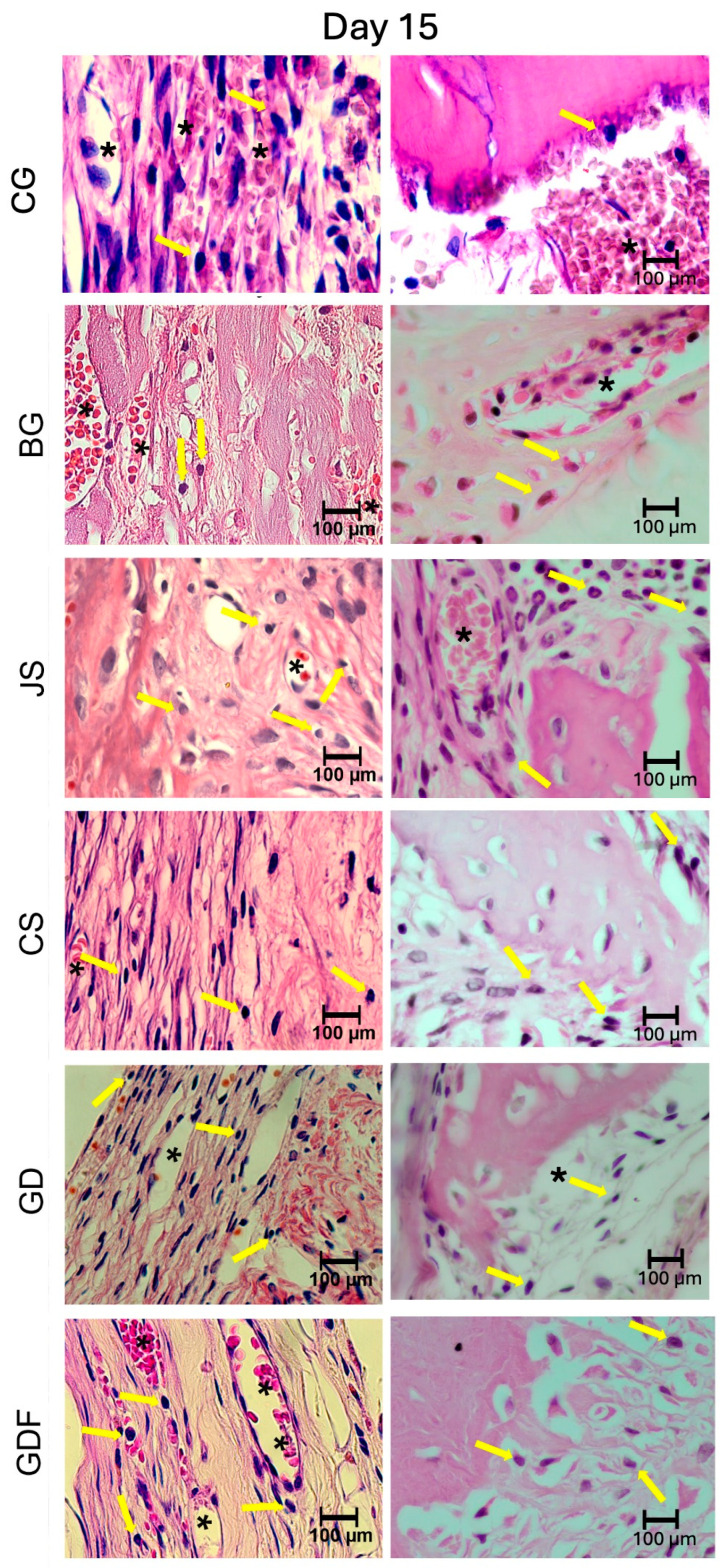
Photomicrographs at 100× magnification demonstrating the presence of cellular content (inflammatory cells and blood vessels) in the CG, BG, JS, CS, GD, and GDF groups at 15 days at the edges and at the center of the defect. Lymphocytes are indicated by yellow arrows and blood vessels by black asterisks.

**Table 1 biomimetics-09-00431-t001:** Classification of the absorbable membranes.

Membrane Types/ Characteristics	Bio-Gide	Jason	Collprotect	GemDerm	GemDerm Flex
Thickness	4 mm	0.15 mm	0.4 mm	0.2 mm	0.25 mm
Resource	Porcine dermis—collagen types I and III	Pericardium porcino—collagen Types I and III	Porcine dermis—collagen types I and III	Bovine bone cortex—collagen type I	Bovine bone cortex—collagen Type I
Degradation time	2–4 weeks	12–28 weeks	8–12 weeks	6 weeks	8 weeks

## Data Availability

The data presented in this study are available upon request from the corresponding author.

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
