# Peer review of "Inflammatory Profile of Different Absorbable Membranes Used for Bone Regeneration: An In Vivo Study"

_biomimetics, 2024, doi:10.3390/biomimetics9070431_

Round 1

Reviewer 1 Report

Comments and Suggestions for Authors

This article compared the inflammation profiles and new-vessel formation ability of five collagen membranes in rat calvaria critical-size defects. It found that Bio-Gide® (BG) and GemDerm Flex® 66 (GDF) had similar inflammatory responses at 7 days and stimulated the angiogenesis. But the manuscript is not well-organized and rigorous.

1.     The full names of BG, JS, CS, GD and GDF in the abstract, and “GBR” in page 1 should be given.

2.      Was the dosage of ketamine hydrochloride along with xylazine in Section 2.2 too high?

3.     HE photomicrographs of the defect edges should be displayed.

4.     Ref [6] didn’t mention in the article. Ref [14, 15, 16] followed Ref [3-5]. The sequence of references displayed in the introduction was unordered.

5.     The thickness, resource and crosslinking method of the membranes should be displayed in a table, therefore the discussion would be more concise.

6.     Which factor played the essential role in inflammatory responses and angiogenesis? Degradation rate, thickness, species or cross-linking methods?

7. The control group (defects with no treatment) was not given. HE images didn't point out the residual collagen membranes. The overall images of the slides should be given.

Comments on the Quality of English Language

Good

Author Response

Dear reviewer, all your considerations have been carefully reviewed and we hope that the questions and improvements have been made correctly.

Reviewer 2 Report

Comments and Suggestions for Authors

Dear authors

The article concerns a very current issue, but both its presentation part and the literature introduction require thorough consideration and presentation.

Line 80 - all tools are sterilized to avoid septic infection in subsequent treatments.

Line 109 – why was this program selected for research? I do not question the credibility and quality of the tool used (because it is a good program). Still, there are many other photo-processing tools with a widely advanced image transformation module commonly used by many researchers worldwide. When using a commercial microscope, the manufacturer probably also provided appropriate software - is the scope of the scientific thesis satisfactory to the authors?

The presented statistical data are quite briefly described - in fact, only the results from the charts are given without a thorough analysis (including comparative analysis) - I suggest you consider the description of the results obtained and a more thorough description of what happened at the cutting site and only refer to the statistical results - preferably with using the text from the discussion section.

I propose to combine both points concerning each case and make a comprehensive description for each analyzed case. The separate breakdown of the statistical results, Photomicrographs, and the later Discussion chapter interprets the research results extremely difficult.

When presenting data in statistical terms, the research group, reference group, and the results obtained should be carefully presented. Wouldn't it be better to present specific research results as the empirical part of the work and only then refer to statistical calculations? With such a small research group, this would be very clear. The results of static tests would be an excellent complement and justify the reason for this type of presentation of results (line 115). Many statistical models are perfect for presenting the above-mentioned issue.

Only 2 publications out of 32 included in the literature review concern the current state of knowledge from the last 5 years. More than 40% of the cited publications are works older than 20 years, so it can be suspected that during this period the progress in the development of the field of knowledge indicated in the publications is at a completely different scientific level. The literature review should indicate current global trends in the development of a given issue and be the basis for defining the research problem that has been solved (constitutes the basis for scientific considerations) in the publication.

To sum up.

A very interesting article concerns current scientific issues with great application possibilities. The literature review needs to be improved, because the proposed publications, although very good in terms of content, differ from the current state of knowledge and reports on current research trends.

Please clearly indicate the author's contribution to the development of a given field. It must be clearly described as to your original and creative contribution to the development of the issue. Testing commercial materials available on the market is not sufficient because companies introducing a product to the market must perform appropriate tests.

Author Response

Dear reviewer, thank you for your considerations, I hope it was up to the mark and we are willing to make further improvements if necessary.

Round 2

Reviewer 1 Report

Comments and Suggestions for Authors

The manuscript should be modified.

1.     The control group (defects with no treatment) was not decribed in "Materials and Methods". Is CG in Figure 7 and 8 was the control group?

2.     The residual collagen membranes was still not pointed out in HE images. Is there overall images of the slides?

Comments on the Quality of English Language

Good

Author Response

Dear reviewer, thank you for your consideration and we remain at your disposal for future questions and improvements.

1. The control group (defects with no treatment) was not decribed in "Materials and Methods". Is

CG in Figure 7 and 8 was the control group?

Response: The abstract and Materials and Methods were improved with the CG (clot group)

information. Yes, the figure 7 was divided in figure 7 and 8 to include the the photomicrophs that

were request in the first review.

2. The residual collagen membranes was still not pointed out in HE images. Is there overall

images of the slides?

Response: Yes, there is overall images of the slides that we can add to the manuscript. Initially,

they were not added because they did not provide information about the primary outcome,

which is the assessment of the inflammatory profile, which can only be observed at higher

magnifications. However, if necessary, we can include the images at a lower magnification.

Reviewer 2 Report

Comments and Suggestions for Authors

The paper submitted for review is an analogous version of the article submitted to the journal. The only significant change is the addition of one author and 3 references that do not significantly affect the substantive value of the submitted work. The changes introduced are cosmetic and do not affect the substantive value of the work.

Author Response

Dear reviewer, thank you for your consideration and we remain at your disposal for future improvements.

The paper submitted for review is an analogous version of the article submitted to the journal.

The only significant change is the addition of one author and 3 references that do not

significantly affect the substantive value of the submitted work. The changes introduced are

cosmetic and do not affect the substantive value of the work.

Response: Dear reviewer, we made the requested changes so that we could respond and pay

attention to the demands made by you and the other reviewer, some of which were conflicting. If

possible, we asked you to point out points to be rewritten and improved so that our manuscript

can move forward. The references have been improved and updated. Thank you for your attention and willingness to improve our study

Round 3

Reviewer 2 Report

Comments and Suggestions for Authors

Dear Authors,

Work submitted for re-evaluation has not been corrected in the major way. Introduced changes were cosmetic and do not bring too much substantive value. Detailed comments present in the previous review were not taken into consideration.

Readability, for example of photos, was higher in the previously sent version of the article, the reduction of photo sizes is making the readability worse.

The literature part was not completed and positions lit 6-8 are as archival as the literature in the previous version of the article. Using articles from different authors does not change that fact – focusing on newer articles (after year 2020) rather than the older ones may be beneficial as it allows locating your research (field, methodology, results) in the current research trends.

In last 5 years, the knowledge in that field has been gradually developing – the topics presented in years 2017-2020 – the research opportunities, results presentation and current know-how are on the completely different level. On the other hand, the research topics published in years 2001 – 2010 are mainly outdated and problems described in them are solved and widely described. Using those articles does not gain any value to your publication.

Author Response

Work submitted for re-evaluation has not been corrected in the major way. Introduced changes were cosmetic and do not bring too much substantive value. Detailed comments present in the previous review were not taken into consideration.

Readability, for example of photos, was higher in the previously sent version of the article, the reduction of photo sizes is making the readability worse.

The literature part was not completed and positions lit 6-8 are as archival as the literature in the previous version of the article. Using articles from different authors does not change that fact – focusing on newer articles (after year 2020) rather than the older ones may be beneficial as it allows locating your research (field, methodology, results) in the current research trends.

In last 5 years, the knowledge in that field has been gradually developing – the topics presented in years 2017-2020 – the research opportunities, results presentation and current know-how are on the completely different level. On the other hand, the research topics published in years 2001 – 2010 are mainly outdated and problems described in them are solved and widely described. Using those articles does not gain any value to your publication.

Response: We have made changes to the article that we now believe are effective. The images were changed to a better resolution and the references were updated, which allowed some parts of the work to be rewritten as it was updated.